# Meningeal contrast enhancement in multiple sclerosis: Assessment of field strength, acquisition delay, and clinical relevance

Daniel M. Harrison [1,2]*, Yohance M. Allette[1,2,3], Yuxin Zeng[1], Amanda Cohen[1], Shishir Dahal[1], Seongjin Choi[1], Jiachen Zhuo[4], Jun Hua[5,6]

1 Department of Neurology, University of Maryland School of Medicine, Baltimore, Maryland, United States of America, 2 Department of Neurology, Baltimore VA Medical Center, VA Maryland Healthcare System, Baltimore, Maryland, United States of America, 3 Department of Neurology, Penn State University–Hershey School of Medicine, Hershey, Pennsylvania, United States of America, 4 Department of Diagnostic Radiology and Nuclear Medicine, University of Maryland School of Medicine, Baltimore, Maryland, United States of America, 5 Department of Radiology and Radiological Sciences, Johns Hopkins University School of Medicine, Baltimore, Maryland, United States of America, 6 F. M. Kirby Research Center for Functional Brain Imaging, Kennedy Krieger Institute, Baltimore, Maryland, United States of America

* dharrison@som.umaryland.edu

**Data Availability Statement:** Original data, including MRI images and clinical database values cannot be shared publicly because of protected health information of human subjects. However,

## Abstract

### Background/Purpose

Leptomeningeal enhancement (LME) on post-contrast FLAIR is described as a potential biomarker of meningeal inflammation in multiple sclerosis (MS). Here we report an assessment of the impact of MRI field strength and acquisition timing on meningeal contrast enhancement (MCE).

### Methods

This was a cross-sectional, observational study of 95 participants with MS and 17 healthy controls (HC) subjects. Each participant underwent an MRI of the brain on both a 7 Tesla (7T) and 3 Tesla (3T) MRI scanner. 7T protocols included a FLAIR image before, soon after (Gd+ Early 7T FLAIR), and 23 minutes after gadolinium (Gd+ Delayed 7T FLAIR). 3T protocol included FLAIR before and 21 minutes after gadolinium (Gd+ Delayed 3T FLAIR).

### Results

LME was seen in 23.3% of participants with MS on Gd+ Delayed 3T FLAIR, 47.4% on Gd+ Early 7T FLAIR (p = 0.002) and 57.9% on Gd+ Delayed 7T FLAIR (p < 0.001 and p = 0.008, respectively). The count and volume of LME, leptomeningeal and paravascular enhancement (LMPE), and paravascular and dural enhancement (PDE) were all highest for Gd+ Delayed 7T FLAIR and lowest for Gd+ Delayed 3T FLAIR. Non-significant trends were seen for higher proportion, counts, and volumes for LME and PDE in MS compared to HCs. The rate of LMPE was different between MS and HCs on Gd+ Delayed 7T FLAIR (98.9% vs 82.4%, p = 0.003). MS participants with LME on Gd+ Delayed 7T FLAIR were older (47.6 (10.6) years) than those without (42.0 (9.7), p = 0.008).

anonymized, limited data sets can be made available through the University of Maryland, Baltimore for researchers who meet the criteria for access to confidential data.

**Funding:** Data acquisition was funded by grants from the NIH (1R01NS104403-01) and Roche-Genentech. Funders had no direct control over any aspects of the study. The funders had no role in study design, data collection and analysis, decision to publish, or preparation of the manuscript.

**Competing interests:** Dr. Harrison has received research funding from EMD-Serono and Roche-Genentech and royalties/consulting fees from the American College of Physicians, Horizon Therapeutics, TG Therapeutics, EMD-Serono, and UpToDate Inc. Drs. Allette, Zeng, Cohen, Dahal, Choi, Zhuo and Hua are without disclosures. These relationships do not alter our adherence to PLOS ONE policies on sharing data and materials.

## Conclusion

7T MRI and a delay after contrast injection increased sensitivity for all forms of MCE. However, the lack of difference between groups for LME and its association with age calls into question its relevance as a biomarker of meningeal inflammation in MS.

## Introduction

Meningeal inflammation is an increasingly recognized contributor to the pathology of multiple sclerosis (MS) [1–3]. Histopathology reveals that most people with MS have widespread inflammatory infiltrates in the meninges [3], and approximately one-half of those with secondary progressive MS (SPMS) have focal lymphoid follicles in the meninges [4]. The recognition of this aspect of MS at autopsy led to a focus on the development of potential *in vivo* neuroimaging markers of meningeal inflammation. One such candidate is meningeal contrast enhancement (MCE) with gadolinium-based contrast agents on Fluid-Attenuated Inversion Recovery (FLAIR) MRI. Although initial attempts to visualize MCE on T1 or FLAIR in MS did not reveal significant findings [5], later work showed that higher resolution 3-dimensional (3D) FLAIR MRI and a delay after contrast administration improved sensitivity for this finding [6]. The visualization of leptomeningeal enhancement (LME) is now described in multiple MS cohorts–studies which also suggest a relationship between LME and cortical gray matter pathology and disability [6–8]. However, although meningeal inflammation is seen in nearly all MS patients at autopsy, most MRI studies only describe LME rates around 21% in MS [9], suggesting the need for a more sensitive biomarker.

Evaluations of LME on ultra-high field 7 Tesla (7T) MRI by this research group and others has revealed far greater rates of LME, typically reported between 66–90% in MS studies at 7T [9–13]. These studies also describe other enhancement patterns, including contrast deposition around large cortical veins, near venous sinuses, and attached to dural tissues [14], which also have been confirmed at 3T [15]. The work performed at 7T suggests that this technology may provide an opportunity to evaluate meningeal inflammation at a level that is much closer to histopathologic descriptions. Further, the initial data from lower field MRI studies suggest initiation of the FLAIR sequence after a post-gadolinium injection delay also improves sensitivity [6, 16]. However, until direct head-to-head evaluations between various field strengths and timing of sequences are performed in the same cohort, the most advantageous method cannot be confirmed.

In this study, we aimed to perform an evaluation of MCE in MS, comparing the rates, count, and volume of various patterns of MCE at different magnetic field strengths and acquisition delay. We also aimed to confirm the clinical relevance of MCE in MS by comparing findings between those with MS and a control group.

## Methods

### Standard protocol approvals, registrations, and patient consents

Protocols were reviewed and approved by the Institutional Review Board (IRB) at the University of Maryland, Baltimore and at the Johns Hopkins University/Kennedy Krieger Institute. All participants reviewed and signed written informed consent documents. Participants with MS were recruited at clinical visits with the investigators. Participants with MS met 2017 revised McDonald Criteria [17] and contained the following phenotypes: relapsing remitting

MS (RRMS), SPMS, and primary progressive MS (PPMS). Healthy controls (HCs), approximately matched to the MS group for age and sex, were recruited through advertisements. HCs were required to have no prior history of any neurological illness. The data reported here includes recruitment that occurred from October 2019 through September 2022.

## Study visits

Demographic information and clinical data were collected through survey-based measures at the study visit and chart review. Expanded Disability Status Scale (EDSS) was scored based on a neurological examination. The study coordinator collected additional measures: Timed 25-foot Walk Test (T25W), Symbol Digit Modality Test (SDMT), Paced Auditory Serial Addition Test (PASAT), and the 9-Hole Peg Test (9HPT). Average values were calculated and utilized for the study regarding tests involving multiple trials, such as the 9HPT and T25W.

## MRI acquisition

Data for this analysis was collated from two separately funded studies with identical clinical data collection and identical 7T imaging protocols. In the larger of the two studies, each participant underwent a 7T MRI and a 3T MRI of the brain a minimum of 1 week apart, but within 30 days of each other. This timing was chosen to ensure full clearance of gadolinium between scans yet performance of both scans within a close enough time gap to not allow for new disease activity to occur. Only 7T MRI scans were acquired in the smaller of the two protocols. 7T MRIs were acquired on a 7T Phillips Achieva scanner (Phillips Healthcare, Best, The Netherlands) with an 8-channel, multi-transmit and 32 channel receive head coil (NovaMedical, Wilmington, MA USA). Acquisitions included a 3D FLAIR sequence (0.49 x 0.49 x 0.5 mm$^3$ resolution) that was collected before contrast and then was initiated again as the first sequence after contrast (gadoteridol 0.1mmol/kg) injection (termed Gd+ Early 7T FLAIR) and initiated again approximately 23 minutes after contrast injection (termed Gd+ Delayed 7T FLAIR). Note that the 7T scanner used is not equipped with a power injector, so, actual sequence initiation likely 1–2 minutes after injection, given time for technician to manipulate patient and machinery. Detailed sequence parameters for this and other sequences are found in S1 Appendix. Magnetization-prepared 2 Rapid Acquisition Gradient Echo (MP2RAGE) images were also acquired, pre- and post-contrast administration at 0.69 x 0.69 x 0.7 mm$^3$ resolution.

3T MRIs were acquired on a Siemens Prisma 3T Whole Body MRI system equipped with a 64 channel receive-only head neck coil. Gadobutrol was used as the contrast agent for 3T MRI acquisition, with the same dosing concentration for the 7T scans. A 3D FLAIR (0.5 x 0.5 x 1.0 mm$^3$ resolution) was acquired before administration of gadobutrol and was initiated again approximately 21 minutes after injection (termed Gd+ Delayed 3T FLAIR). An MP2RAGE was also acquired with 1.0 mm$^3$ isotropic resolution.

The timing of the delayed contrast FLAIR acquisitions was based on prior literature on enhancement kinetics [18] and to provide time for completion of other sequences (including the Gd+ Early 7T FLAIR and post-contrast MP2RAGE) before initiation.

## Image processing

Processing methods have been detailed in our prior work on this subject [10, 11, 13, 14, 19]. Briefly, 7T MRI images underwent post-processing to create MP2RAGE T1-weighted (T1W) images using custom code [20]. This process was performed on-scanner using manufacturer provided algorithms for the 3T scans. Following N4 inhomogeneity correction [21], 7T T1W images were denoised by multiplying this by the second inversion time image in the corresponding MP2RAGE acquisition. The denoising process was not necessary for the 3T images

due to on-scanner processing. Denoised T1W images were used for skull stripping and co-registration. Both the pre- and post-contrast FLAIR images (both 3T and 7T sequences) were co-registered to the denoised pre-contrast T1W images (of the same field strength) following N4 inhomogeneity correction utilizing ANTs (Advanced Normalization Tools, https://github.com/ANTsX/ANTs) [21]. The denoised 3T T1W scans were registered to the equivalent 7T T1W, and the transformation matrix for this registration was applied to the 3T FLAIR images to bring them into the common analysis space. FLAIR subtraction maps were created by subtracting the intensity values of the pre-contrast FLAIR image from the post-contrast image.

## Image analysis

Analysis methods were similar to our prior publications on this topic. An initial reviewer (DMH)) with 15 years of neuroimaging research experience and 5 years of specific expertise in post-contrast FLAIR MCE analysis completed randomized evaluation of images to identify and categorize foci of ME. Images reviewed during this process included the pre-contrast FLAIR, the post-contrast FLAIR, FLAIR subtraction, and pre- and post-contrast T1W. See Fig 1 for examples of this process. Any images with unacceptable quality for this review (i.e. motion artifact, poor CSF signal suppression, etc.) were discarded. Foci of post-contrast

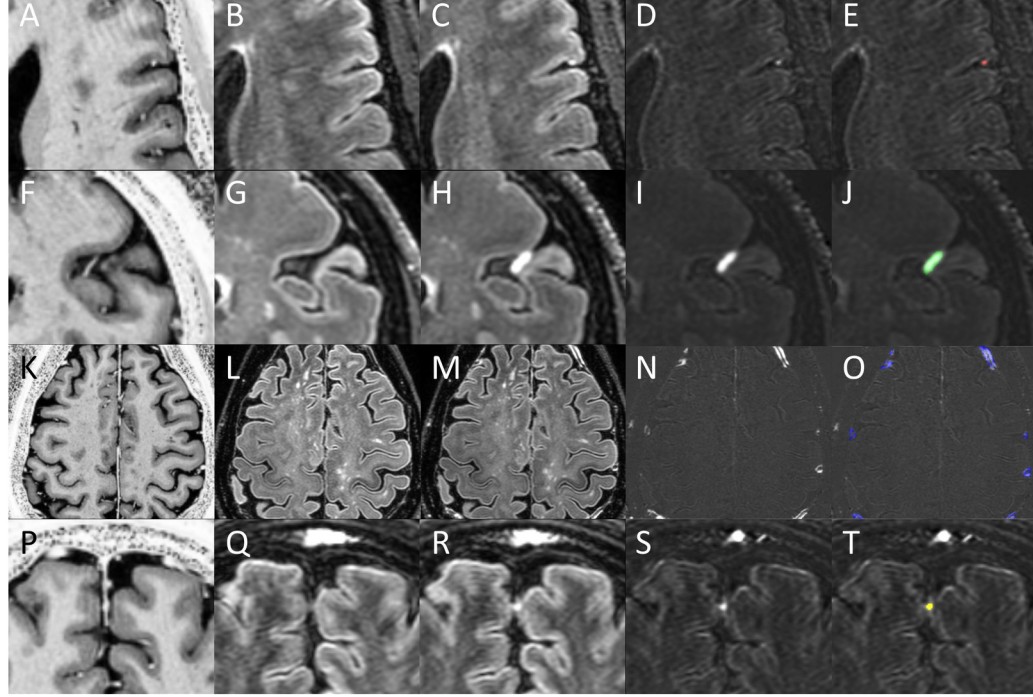

**Fig 1. Identification of meningeal enhancement.** Shown are Gd+ MP2RAGE T1W (A, F, K, P), Gd- 7T FLAIR (B, G, L, Q), Gd+ Delayed 7T FLAIR (C, H, M, R), FLAIR subtraction maps (D, I, N, S), and FLAIR subtraction maps with overlaid enhancement masks (E, J, O, T). Regions of enhancement were first identified on FLAIR subtraction maps, and then confirmed by evaluation on FLAIR and T1W images. Subtypes were then classified, and masks drawn as the focus appeared on the subtraction map. A nodular focus (red mask) is shown on the top row and a spread/fill focus (green mask) is shown on in the 2nd row. Together, these subtypes were classified as leptomeningeal enhancement (LME). Note the proximity of both types of LME foci to meningeal blood vessels on Gd+ T1W. Multiple paravascular foci (blue masks) are shown in the 3rd row. Note the location of enhancements to be immediately external to enhancing meningeal blood vessel lumens on Gd+ T1W. A dural nodule focus (yellow mask) is shown in the 4th row. Note the co-localization of this focus to enhancing structures in the falx cerebri on Gd+ T1W. Paravascular and dural nodules together were termed paravascular and dural enhancement (PDE).

enhancement were initially identified as hyperintense regions on FLAIR subtraction maps. These foci were then confirmed by review of pre- and post-contrast FLAIR to ensure that subtraction hyperintensities were indeed due to lack of signal on pre-contrast and presence of signal on post contrast and were not due to motion artifacts or registration errors. Location and subtype were aided by co-localization of the coordinates on T1W images. Images for Gd + Early 7T FLAIR, Gd+ Delayed 7T FLAIR, and Gd+ Delayed 3T FLAIR were reviewed in separate, unrelated analysis sessions. Labelling of enhancement foci category was similar to our prior publications on this topic [10, 11, 13, 14]. Briefly, "nodular" foci at the pial surface and amorphous areas of contrast in the subarachnoid space ("spread/fill") were categorized as leptomeningeal enhancement (LME). Nodular appearing foci within dura ("dural nodule") and regions of contrast surrounding the walls of large cortical veins and venous sinuses ("paravascular") were not included in the LME definition as described by Absinta et al [6], and were thus grouped as paravascular/dural enhancement (PDE). We also explored an alternate grouping definition, leptomeningeal and paravascular enhancement (LMPE), which included nodular, spread/fill, and paravascular, given that cortical veins traverse the subarachnoid space before merging with dura [22]. A team of additional independent reviewers (SD, YMA, AHC) re-reviewed the initial demarcations of the initial reviewer (DMH) for confirmation of foci presence and sub-type labelling. Suggested changes (removal of foci, addition of foci, changing of sub-type labelling) by these reviewers were reviewed by the initial reviewer to achieve consensus. After consensus, the additional reviewers created three-dimensional maps of the identified foci with semi-automated region filling tools in MIPAV (Version 11.0.3), from which volumes were derived. Sample masks are shown in Fig 1.

### Statistical analysis

Statistical analyses were completed in IBM SPSS (version 29), Stata 10.1 IC, or R (version 4.2.2). Evaluation of data distribution was completed with both the Kolmogorov-Smirnov and Shapiro-Wilk testing. Results revealed non-normal distributions for most data and thus non-parametric analysis methods were used for group comparisons by the independent samples Kruskal-Wallis test for multi-group comparisons and Wilcoxon rank sum testing for binary group comparisons. The few exceptions that qualified for use of Student's t-test are highlighted as such in Tables and text. Chi-square test or Fisher's exact test was used for comparison of proportions. P-values for group comparisons were corrected by the Bonferroni method and significance for adjusted p-values was set to $p < 0.05$. Adjusted p-values are reported in the text.

Relationships between imaging findings and demographic/clinical characteristics were also evaluated by regression modeling. Volumes were assessed with a linear regression model, counts were assessed with a negative binomial regression model, and logistic regression was used for binary meningeal enhancement outcomes. All models were initially tested for overdispersion and goodness of fit, and only those passing these evaluations with $p < 0.05$ were evaluated and reported. Negative binomial model assumptions were also assessed visually. All models were adjusted for co-variates of significance, such as age, sex, and disease duration.

## Results

### Cohort

Data from 95 participants with MS, 76 (80%) of which had RRMS, were used, along with 17 HCs (Table 1). Of these, 60 (63.2%) MS participants and 16 (94.1%) HCs had paired 3T scans available for comparison to 7T. MS participants had a median disease duration of 11.4 (0.4–47.8) years and had moderate levels of disability, with a median EDSS score of 2.5 (0–6.5). The

**Table 1. Demographic and clinical characteristics of study participants.**

| | | MS | HC |
|---|---|---|---|
| | | **N = 95** | **N = 17** |
| Mean Age (SD), years | | 45.2 (10.5) | 42.2 (13.7) |
| Female sex (%) | | 66 (69.5) | 11 (64.7) |
| Median Disease Duration [min-max], years | | 11.4 [0.4–47.8] | n/a |
| Median Number of Relapses in Past 12 months [min-max] | | 0 [0–3] | n/a |
| Relapses Treated with Corticosteroids in Past 30 days (%) | | 4 (4.2) | n/a |
| Clinical phenotype (%) | | | n/a |
| Relapsing-remitting | | 76 (80) | |
| Secondary Progressive | | 8 (8.4) | |
| Primary Progressive | | 11 (11.6) | |
| Disease Modifying Therapy (%) | None | 19 (20) | n/a |
| | Interferon | 2 (2.1) | n/a |
| | Glatiramer Acetate | 7 (7.4) | n/a |
| | Natalizumab | 12 (12.6) | n/a |
| | Teriflunomide | 2 (2.1) | n/a |
| | S1P Modulator | 9 (9.5) | n/a |
| | Fumarates | 13 (13.7) | n/a |
| | Anti-CD-20 Therapy | 30 (31.6) | n/a |
| | Corticosteroids | 0 | n/a |
| | Other | 1 (1.1) | n/a |
| Disability Scales | Median EDSS | 2.5 [0–6.5] | n/a |
| | [min-max] | | |
| | Median 9HPT | 22.6 [15.6–425.7] | 19.9 |
| | [min-max] | | [16.8–29.5] |
| | Median T25W | 5.3[*] | 4.3[*] |
| | [min-max] | [3.2–161] | [3.6–5.3] |
| | Mean SDMT (SD) | 53.9 (15.1) | 56.8 (11.2) |
| | Mean PASAT (SD) | 45 [0–60] | 46 [15–60] |

HC = healthy control, MS = multiple sclerosis, SD = standard deviation. EDSS = Expanded Disability Status Scale, 9HPT = Nine Hole Peg Test (in seconds), T25W = Timed 25-foot walk (in seconds), SDMT = Symbol Digit Modalities Test (# correct), PASAT (# correct).

"[*]" = $p < 0.05$ for comparison of HC and MS

Kruskall-Wallis or Wilcoxon rank sum tests for values showing medians, t-test for those showing means, and Chi-square or Fisher's exact test for those showing proportions. Note, statistical tests comparing groups for disease modifying therapies not performed due to lack of clinical relevance.

cohort was relatively stable, with 73 (76.9%) participants having no relapses in the past year and 92 (96.8%) having no relapses in the past 30 days. Eighty percent of the MS participants were on disease modifying therapy (DMT), with anti-CD20 therapy (31.6%) constituting the most prevalent treatment.

## Qualitative observations during image analysis

All the previously described MCE subtypes were observed on review of gadolinium-enhanced FLAIR images both at 3T and 7T. Examples are seen in Fig 1. Both nodular and spread/fill type LME were almost always found at the same location as, or in the near vicinity of, small veins that were visualized in the subarachnoid space only on post-contrast T1W images.

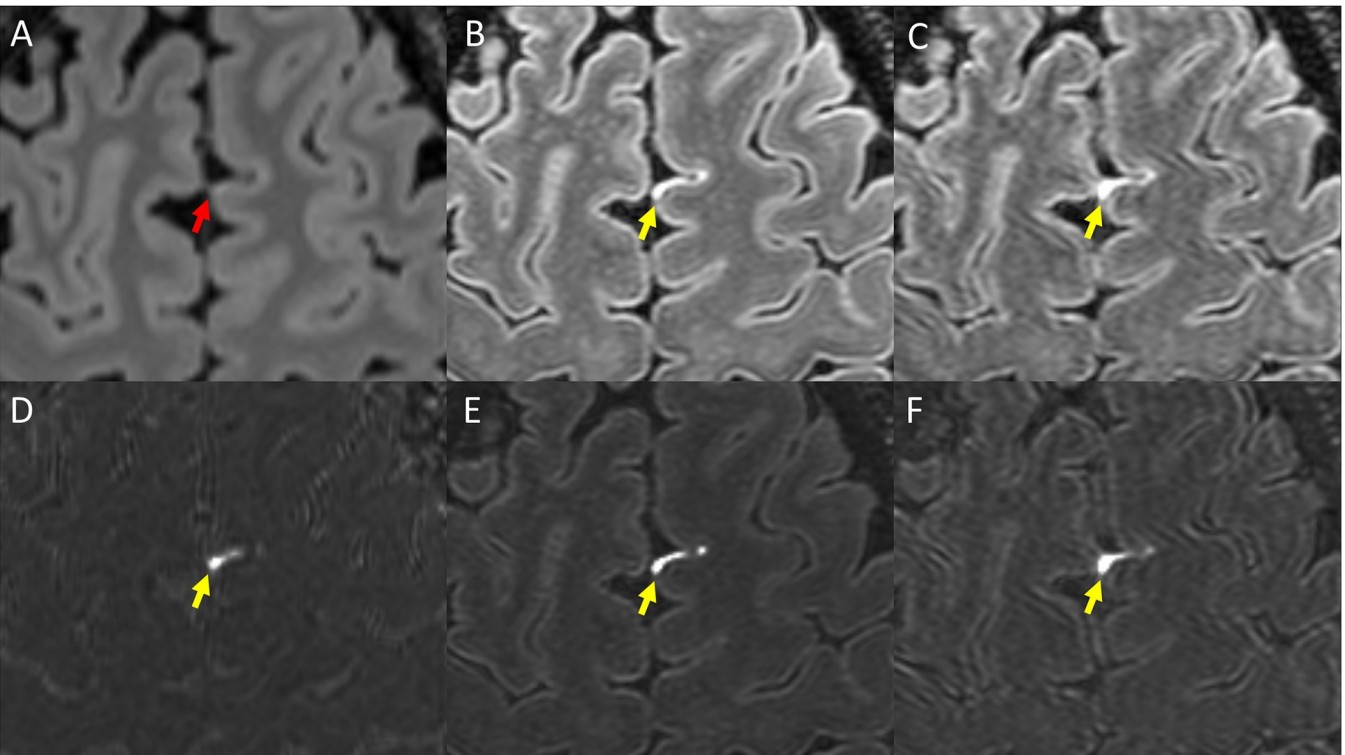

**Fig 2. Example of LME as visualized on various Gd+ imaging protocols.** Shown is Gd+ Delayed 3T FLAIR (A), Gd+ Early 7T FLAIR (B), and Gd+ Delayed 7T FLAIR (C). Also shown are subtraction maps created by subtraction of A from pre-Gd 3T FLAIR (D) and B and C subtracted from pre-Gd 7T FLAIR (E and F, respectively). Yellow arrows indicate a focus of LME readily visible, but which is barely visible on Gd+ Delayed 3T FLAIR (red arrow). Note that although the focus is seen on Gd+ Early 7T FLAIR (B, E), it is thicker/more prominent on Gd+ Delayed 7T FLAIR (C, F).

Paravascular enhancement was always found just outside the boundary of larger cortical veins that were visible on post-contrast T1W images–veins that were either minimally seen on pre-contrast T1W or not seen at all. These foci often tracked along the region just outside of these blood vessels for large portions of their course to the dura and/or dural sinuses. Dural nodules were frequently seen along the falx cerebri in the same location as enhancing structures on T1W, at or near the outer edge of dural sinuses, or in other regions of dura.

The size and intensity of enhancing foci were generally increased on 7T scans compared to 3T. Examples are shown in Figs 2 and 3. As seen in these figures, LME and PDE foci at times were barely visible on Gd+ Delayed 3T FLAIR, more visible on Gd+ Early 7T FLAIR, and most prominent on Gd+ Late 7T FLAIR. Instances of LME or PDE foci not seen at all on Gd + Delayed 3T FLAIR and/or Gd+ Early 7T FLAIR but visible on Gd+ Delayed 7T FLAIR were also noted.

LME and PDE foci were seen in both cohorts (MS, HC). When present, LME foci seen in HCs had a visual appearance that was indistinguishable from participants with MS (Fig 4). The same was seen for PDE.

## Imaging sequence comparison

Detailed comparisons are seen in Table 2 and in S1 Table. LME was seen in 23.3% of participants with MS on Gd+ Delayed 3T FLAIR, compared to 47.4% for Gd+ Early 7T FLAIR (p = 0.002) and 57.9% for Gd+ Delayed 7T FLAIR (p < 0.001 and p = 0.008, respectively). The number of foci of LME seen in the MS cohort was lowest for Gd+ Delayed 3T FLAIR

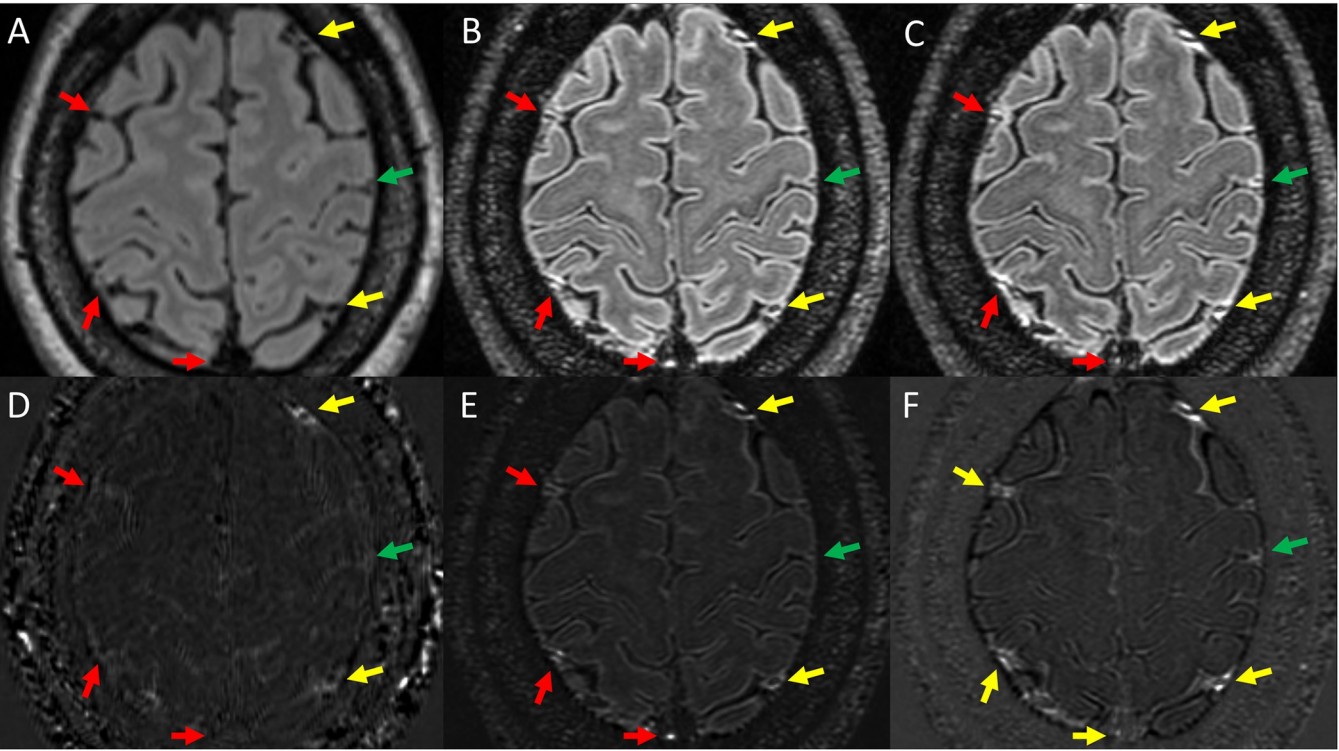

**Fig 3. Examples of PDE as visualized on various Gd+ imaging protocols.** Shown is Gd+ Delayed 3T FLAIR (A), Gd+ Early 7T FLAIR (B), and Gd+ Delayed 7T FLAIR (C). Also shown are subtraction maps created by subtraction of A from pre-Gd 3T FLAIR (D) and B and C subtracted from pre-Gd 7T FLAIR (E and F, respectively). Yellow arrows indicate foci of PDE seen on all 3 protocols. Red arrows indicate regions where PDE was seen on both 7T protocols but could not be seen on Gd+ Delayed 3T FLAIR. The green arrow indicates a region where PDE was seen on Gd+ Delayed 7T FLAIR, but could not be seen on Gd + FLAIR or Gd+ Early 7T FLAIR.

(p = 0.028 vs Gd+ Early 7T FLAIR, p = 0.018 vs Gd+ Delayed 7T FLAIR) and highest for Gd + Delayed 7T FLAIR (p = 0.036 vs Gd+ Early 7T FLAIR). LME volume was increased in Gd + Delayed 7T FLAIR compared to Gd+ Delayed 3T FLAIR (p = 0.005) but was not significantly increased when compared to Gd+ Early 7T FLAIR. Similar patterns to the above were seen when looking at nodular and spread/fill LME individually, with the LME findings mostly driven by the larger and more numerous spread/fill foci (S1 Table). The variations in LMPE across the 3T and 7T sequences appeared in a similar pattern to LME.

PDE was seen in 65% of MS participants on Gd+ Delayed 3T FLAIR, compared to 86.3% on Gd+ Early 7T FLAIR (p = 0.002) and 96.8% on Gd+ Delayed 7T FLAIR (p < 0.001 and p = 0.005, respectively). The number of PDE foci seen in the MS cohort was smallest on Gd + Delayed 3T FLAIR and highest on Gd+ 7T Delayed FLAIR (p < 0.001 for all comparisons). PDE enhancing volume was also smallest on Gd+ Delayed 3T FLAIR, higher on Gd+ Early 7T FLAIR (p = 0.014), and highest on Gd+ Delayed 7T FLAIR (p < 0.001 for both comparisons). Similar patterns to the above were seen when paravascular and dural foci were evaluated independently, with most of the PDE findings likely deriving from the more numerous and larger paravacular foci (S1 Table).

## Group comparisons

Although there were trends towards a larger volume and a higher number of LME foci in MS participants (Table 2), particularly for Gd+ Delayed 7T FLAIR, none of the comparisons

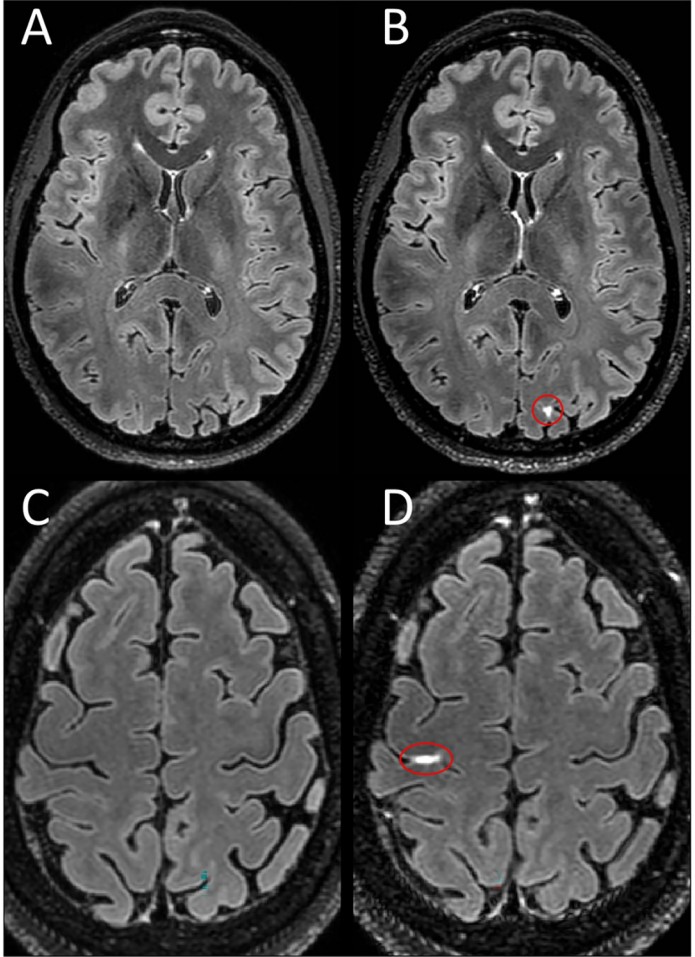

**Fig 4. LME in healthy controls.** Shown are axial Gd+ Delayed 7T FLAIR images with LME highlighted by red ovals. Examples of LME in healthy controls as seen in a 40-year-old woman (B) and a 47-year-old woman (D) with no known neurological conditions. No leptomeningeal hyperintensity is seen in the same location on pre-contrast images (A,C).

between MS and HCs attained significance. The rate of LMPE, however, was significantly different between MS and HCs on Gd+ Delayed 7T FLAIR (98.9% vs 82.4%, p = 0.003). Furthermore, the number of LMPE foci in MS on Gd+ Early 7T FLAIR (3 (0–19)) was greater than in HCs (1 (0–8), p = 0.014).

The proportion of participants with PDE was not significantly different between the MS and HC groups. PDE volume trended towards elevation in MS (220.9 (0–4855.1) mm$^3$) compared to HCs (85.7 (0–728.5) mm$^3$ on Gd+ Early 7T FLAIR but failed to attain significance after p-value adjustment (p = 0.076). PDE count was consistently elevated in MS compared to HCs for all 3 imaging protocols (Table 2), but these differences did not attain statistical significance. Results of group comparisons for individual foci subtypes are seen in S1 Table.

### Associations with cohort characteristics and disability

MS participants with LME on Gd+ Delayed 7T FLAIR were older (47.6 (10.6) years) than those without LME (42.0 (9.7) years, p = 0.008) (Table 3). MS participants with LMPE on Gd

**Table 2. Group comparisons for ME volume, counts, and presence.**

| | | MS | | | HC | | |
|---|---|---|---|---|---|---|---|
| | | Volume (mm³) | Count | ME Present (%) | Volume (mm³) | Count | ME Present (%) |
| LME | Gd+ Delayed 3T FLAIR | 9.7 (23.4) | 0.43 (0.9) | 14[b,c] (23.3) | 25.4 (59.8) | 0.5 (0.7) | 6 (37.5) |
| | | 0[c] [0–109] | 0[b,c] [0–3] | | 0 [0–218] | 0 [0–2] | |
| | Gd+ Early 7T FLAIR | 125 (888.2) | 0.9 (1.4) | 45[b,d] (47.4) | 28.4 (49.6) | 0.5 (0.7) | 6 (35.3) |
| | | 0 [0–8623.4] | 0[b,d] [0–9] | | 0 [0–137.9] | 0 [0–2] | |
| | Gd+ Delayed 7T FLAIR | 105.2 (494.5) | 1.4 (1.9) | 55[c,d] (57.9) | 43.9 (106.4) | 1 (1.7) | 8 (47.1) |
| | | 6.4[c] [0–3780.7] | 1[c,d] [0–10] | | 0 [0–420.2] | 0 [0–6] | |
| LMPE | Gd+ Delayed 3T FLAIR | 429.6 (985.0) | 2.6 (3.7) | 38[b,c] (63.3) | 348.6 (745.8) | 2.3 (3.3) | 10 (62.5) |
| | | 66.3[b,c] [0–6505.3] | 1.5[b,c] [0–19] | | 20.5 [0–2858.4] | 1 [0–11] | |
| | Gd+ Early 7T FLAIR | 768.2 (1381.5) | 4.28 (4.2) | 79[b,d] (83.2) | 172.3 (251.7) | 1.8 (2.2) | 11 (64.7) |
| | | 228.4[b,d] [0–9919.9} | 3[a,b,d] [0–19] | | 40.7[d] [0–866.4] | 1[a,d] [0–8] | |
| | Gd+ Delayed 7T FLAIR | 1259.0 (2086.1) | 8.1 (6.6) | 94[a,c,d] (98.9) | 951.5 (1064.7) | 5.6 (4.7) | 14[a] (82.4) |
| | | 600.7[c,d] [0–18112.0] | 6[c,d] [0–35] | | 509.3[d] [0–3424.0] | 5[d] [0–16] | |
| PDE | Gd+ Delayed 3T FLAIR | 435.4 (984.8) | 2.8 (3.8) | 39[b,c] (65) | 332.6 (731.9) | 2.3 (3.3) | 8[c] (50) |
| | | 69.3[b,c] [0–6505] | 2[b,c] [0–19] | | 8.2 [0–2867] | 0.5[c] [0–11] | |
| | Gd+ Early 7T FLAIR | 675.7 (999.7) | 4.4 (4.2) | 82[b,d] (86.3) | 169.8 (214.8) | 2.2 (2.3) | 11 (64.7) |
| | | 220.9[b,d] [0–4855.1] | 3[b,d] [0–21] | | 85.7[d] [0–728.5] | 1[d] [0–6] | |
| | Gd+ Delayed 7T FLAIR | 1182.4 (2054.6) | 8.6 (6.5) | 92[c,d] (96.8) | 924.2 (1021.8) | 5.6 (4.5) | 15[c] (88.2) |
| | | 618.6[c,d] [0–18134.2] | 7[c,d] [0–36] | | 605.7[d] [0–3270.8] | 5[c,d] [0–15] | |

HC = healthy control, MS = multiple sclerosis, LME = leptomeningeal enhancement, LMPE = leptomeningeal and paravascular enhancement, PDE = paravascular/dural enhancement, SD = standard deviation.

Values shown are mean (SD) and median [minimum-maximum]. Mean shown for informational purposes, but statistical tests performed using non-parametric testing, and thus statistical significance to be shown for median values.

"a" = p < 0.05 for comparison of HC and MS

"b" = p < 0.05 for comparison between Gd+ Delayed 3T FLAIR and Gd+ Early 7T FLAIR

"c" = p < 0.05 for comparison between Gd+ Delayed 3T FLAIR and Gd+ Delayed 7T FLAIR

"d" = p < 0.05 for comparison of Gd+ Early 7T FLAIR and Gd+ Delayed 7T FLAIR.

+ Early 7T FLAIR were older (46.4 (10.5) years) than those without LMPE (39.6 (9.0), p = 0.014) (S2 Table). Similar trends were seen for other methods and for PDE but were not significant. No significant differences in sex, disease duration, or any of the disability scales were seen in comparing those with/without LME, LMPE, or PDE for any of the MRI methods in this study. Regression modeling failed goodness of fit testing for most models. The remaining models found only minimal associations between any form of MCE and disability. PASAT scores were associated with dural nodule count (estimate -0.02 (-0.04–1.65), p = 0.007) and volume (estimate -1.54 (-2.57 - -0.50), p = 0.004) on Gd+ 7T Early FLAIR. 9HPT was associated with both paravascular volume (estimate 6.23 (2.02–10.45), p = 0.004), PDE volume (estimate 6.11 (1.89–10.33), p = 0.005), and LMPE volume (estimate 5.84 (0.02–11.66, p = 0.049) on Gd+ Early 7T FLAIR. No other significant associations were seen.

The findings of LME in HCs were further explored by evaluations of the impact of the age of participants. HCs with LME on Gd+ Early 7T FLAIR were older (53 years (36–64)) than those without (33 years (24–58), p = 0.027). A similar, non-significant trend was seen for HCs with LME on Gd+ Delayed 3T FLAIR (43.5 years (28–64) compared to 35.5 years (24–60), p = 0.55)) and no difference for Gd+ Delayed 7T FLAIR (38 years (25–64) compared to 42 years (24–60), p = 0.96)).

**Table 3. Demographic and clinical characteristics by presence of meningeal enhancement in the MS cohort.**

| | | LME | | | | | | PDE | | | | | |
|---|---|---|---|---|---|---|---|---|---|---|---|---|---|
| | | Gd+ 3T LME + N = 14 | Gd+ 3T LME— N = 46 | Gd + Early 7T LME + N = 45 | Gd + Early 7T LME —N = 50 | Gd + Delayed 7T LME + N = 55 | Gd + Delayed 7T LME— N = 40 | Gd+ 3T PDE + N = 39 | Gd+ 3T PDE— N = 21 | Gd + Early 7T PDE + N = 82 | Gd + Early 7T PDE —N = 13 | Gd + Delayed 7T PDE + N = 92 | Gd + Delayed 7T PDE— N = 3 |
| Mean Age (SD) | | 47.21 (10.8) | 44.39 (9.93) | 45.8 (10.0) | 44.7 (11.0) | 47.6* (10.6) | 42.0* (9.7) | 46.6 (10.0) | 42.1 (10.0) | 45.7 (10.6) | 42.7 (10.4) | 45.3 (10.4) | 42.7 (16.2) |
| Median Disease Duration [min-max] | | 14.2 [5.3–29.4] | 12.5 [0.4–33.3] | 11.2 [1.7–47.8] | 12.4 [0.4–33.3] | 11.8 [0.4–47.8] | 10.1 [1.5–24.7] | 14.2 [1.8, 33.3] | 11.8 [0.4, 24.2] | 10.7 [1.2–47.8] | 12.6 [0.4, 22.9] | 11.3 [0.4–47.8] | 12.6 [1.8–17.9] |
| Female Gender (%) | | 12 (85.7) | 29 (63.0) | 30 (66.7) | 36 (72.0) | 39 (70.9) | 27 (67.5) | 30 (76.9) | 11 (52.4) | 56 (68.3) | 10 (76.9) | 64 (69.6) | 2 (66.7) |
| Disability Scales | Median EDSS [min-max] | 2.3 [1.5–6.5] | 2.5 [0–6.5] | 2.5 [0–6.5] | 3 [0–6.5] | 2.5 [0–6.5] | 2.5 [0–6.5] | 2.5 [0–6.5] | 2.5 [0–6.5] | 2.5 [0–6.5] | 2.5 [0–6.5] | 2.5 [0–6.5] | 4 [1.5–6] |
| | Median 9HPT [min-max] | 5.1 [3.9–161] | 4.9 [3.7–161] | 5 [3.2–161] | 5.4 [3.4–161] | 5.5 [3.2–161] | 5.1 [3.4–161] | 4.9 [3.7–161] | 5.2 [3.9–161] | 5.3 [3.2–161] | 5.1 [3.8–24.1] | 5.3 [3.2–161] | 6.4 [4.5–8.2] |
| | Median T25W [mix-max] | 5.1 [3.9–161] | 4.9 [3.7–161] | 5.0 [3.2–161] | 5.4 [3.4–161] | 5.5 [3.2–161] | 5.1 [3.4–161] | 4.9 [3.7, 161] | 5.2 [3.9–161] | 5.3 [3.2–161] | 5.1 [3.8, 24.1] | 5.3 [3.2–161] | 6.4 [4.5–8.2] |
| | Mean SDMT (SD) | 54.6 (15.0) | 53.8 (14.9) | 54.4 (15.5) | 53.4 (14.8) | 54.9 (14.7) | 52.8 (15.5) | 54.0 (16.0) | 54.0 (12.6) | 53.7 (15.1) | 55.5 (15.5) | 54.1 (15.0) | 47.7 (20.0) |
| | Median PASAT [min-max] | 48.5 [0–55] | 48 [0–59] | 46 [0–59] | 44.5 [7–60] | 43 [0–59] | 45.5 [6–60] | 48 [0–59] | 49 [26–56] | 45 [0–60] | 43 [24–56] | 45 [0–60] | 49 [42–56] |

LME = leptomeningeal enhancement, PDE = paravascular/dural enhancement, SD = standard deviation. EDSS = Expanded Disability Status Scale, 9HPT = Nine Hole Peg Test (in seconds), T25W = Timed 25-foot walk (in seconds), SDMT = Symbol Digit Modalities Test (# correct).

Wilcoxon rank sum test was used for all values shown as median, t-test for those shown as mean, and proportions were tested by Chi-square or Fisher's exact test.

"*" = p < 0.05.

## The effect of treatment on MCE

Treatments did not appear to impact the presence/absence of any form of MCE in this cohort. Of the four participants who had received corticosteroids in the 30 days prior to their imaging, two of four had LME and all four had PDE on both their Gd+ Early and Gd+ Delayed 7T FLAIR images. Only one of these participants had a paired 3T scan, on which no LME or PDE was seen.

To evaluate the impact of disease modifying therapy, we split the disease modifying therapies (DMTs) into two categories: high efficacy monoclonal antibodies (natalizumab, alemtuzumab, and anti-CD20 class) and low/moderate efficacy DMTs (i.e. injections and oral treatments). Of the 76 participants on DMT, 42 (55.2%) were on high efficacy monoclonal antibodies. Of the 55 of these subjects with paired 3T scans, 27 (49.1%) were on high efficacy monoclonal antibodies. There were no significant differences seen in the proportion of participants with LME, LMPE, or PDE in those on either treatment category for any of the MRI protocols (S3 Table).

## Discussion

This study provides the largest set of data directly evaluating the impact of acquisition timing and field strength on MCE in MS to date. Our data confirms the hypothesis that visualization of MCE is dependent on both field strength and the timing of acquisition after contrast administration, with a delayed acquisition initiation after contrast administration on 7T FLAIR being the most sensitive of the 3 protocols used in this study. This manifested as higher rates of detection, larger counts, and larger volumes on Gd+ Delayed 7T FLAIR than seen on either Gd+ Delayed 3T FLAIR or Gd+ Early 7T FLAIR. However, the data obtained in this study also calls into question the clinical relevance of the findings of LME, at least per prior definitions, as no significant differences between MS and HCs were seen, along with no major differences in disability testing or evident treatment effects.

The higher rates of detection of MCE on both 7T protocols compared to 3T is consistent with prior data evaluating these findings at those field strengths independently. The detection rate of MCE in most prior MS studies at 7T was 2–3 times greater than in separate studies at 3T [6, 13–15], which was replicated here using both methods in the same participants. Given that most foci of MCE (particularly LME) visualized on 3D-FLAIR are quite small, it is not surprising that 7T methods provide greater sensitivity. The acquired resolution feasible in most 7T 3D-FLAIR protocols is superior to that at 3T, and sub-millimeter resolution is likely necessary to detect small foci and avoid partial volume effects that may obscure subtle foci from visualization. Furthermore, as MCE in MS is not visible on T1W imaging, the concentration of gadolinium released into the meningeal space is likely low, which is further reduced by dilution in CSF. Excellent signal-to-noise ratio is likely necessary to visualize such a subtle finding, which is another advantage to studying this phenomenon on ultra-high field MRI.

The data presented here also supports the concept that the timing of scan initiation post-contrast injection is critical. This finding is not unique to this study. Evaluations of LME in cerebral vascular disease and brain metastases indicate that longer delays after contrast administration significantly increase the prevalence of the finding and alter the visual appearance of regions of enhancement [23–27]. Prior data in MS also suggests that some forms of MCE in MS may have altered intensities depending on the timing of acquisition after contrast injection [6, 28]. This suspicion is confirmed by the results of the comparisons between Gd+ Early 7T FLAIR and Gd+ Delayed 7T FLAIR in this study, which consistently showed higher counts and volumes for all ME sub-types for the delayed version of the protocol. Our findings confirm that a longer contrast delay than what is typically performed in clinical imaging is necessary for high sensitivity to the findings of LME, PDE, and LMPE. However, work is still necessary to determine what the 'ideal' timing for optimization of the study of various forms of MCE in MS may be, or alternately if evaluations of the dynamic changes in enhancement over time post-injection are themselves markers of MS-related pathology.

The high prevalence of MCE in the HC group in this study also calls into question the concept that either LME or PDE are pathologic findings related to MS. Although both findings trended towards higher rates, counts, and volumes in MS, statistical comparison between the three groups revealed only minor differences. A lack of pathologic specificity solely to MS for LME could be expected, given prior work showing prevalent findings of LME in other autoimmune conditions such as neurosarcoidosis, neuromyelitis optica, myelin oligodendrocyte glycoprotein antibody disorder (MOGAD), and Susac syndrome [29–31]. However, the finding that 35.3–47.1% of HCs in this study had evidence of LME (depending on protocol) calls into question the potential link between LME on post-contrast FLAIR and meningeal inflammation. Relatively few studies of LME in neuroinflammatory disease have had substantial control populations, and thus, it is not clear if our finding is an outlier or more indicative of the true

prevalence of LME in the general population. Although the initial publication on LME in MS found no LME in HCs [6], a recent meta-analysis pooled together the few studies utilizing post-contrast FLAIR that included HCs and found a pooled prevalence of 13 (7–20) % [9]. It is not entirely clear why our study revealed rates of LME in HCs at nearly twice that rate, although differences in sampling, ages of participants, and the sensitivity of our 7T methods and use of subtraction imaging [32] may explain these differences. In addition to other studies noting LME in healthy participants, similar findings are described in many non-inflammatory conditions, such as traumatic brain injury, cerebral ischemia, and reversible cerebral vasoconstriction syndrome [9, 25, 29]. Recent studies have even described high rates of LME in memory disorders clinic patients with Alzheimer disease and those with mild cognitive impairment [33], suggesting that neurodegenerative pathology can be associated with this imaging finding. These data and our finding in HCs suggest that the finding of LME on MRI may be on a spectrum with normal physiology but may be increased under certain conditions.

The hypothesis that LME in MS is due to focal meningeal inflammation is not supported by strong evidence. Human data is limited to the evaluation of 3 enhancing foci in 2 autopsy cases in one study [6]. LME is reported to be related to meningeal inflammation in experimental autoimmune encephalomyelitis (EAE) [34, 35], but it is not clear that this phenomena is the same as what is described in MS, as the acute, transient, and diffuse nature of enhancement seen in EAE is not consistent with the more focal and persistent/durable (for years) descriptions of LME in MS [6, 14]. Further, EAE is not a perfect model for human disease, and some aspects of the model are more akin to pathology seen in MOGAD, in which meningeal inflammation can be more profound [30].

Although LME on post-contrast FLAIR has been utilized in the MS literature as a potential surrogate of meningeal inflammation, the same finding is hypothesized to occur by other mechanisms elsewhere. Work in other conditions and HCs suggest the finding may be indicative of glymphatic clearance of gadolinium and/or alterations in central nervous system (CNS) lymphatics and perivascular spaces [25, 36, 37]. Alteration in the blood-brain-barrier (BBB) is a likely common etiology among many conditions, including MS, and the co-localization of most LME foci with blood vessels on T1W in this study supports this notion. The presence of the finding in vascular conditions, neurodegenerative disease, and in normal aging support the idea that LME can occur due to focal or diffuse BBB breakdown, but also demonstrate that meningeal inflammation is not required for this to occur. Thus, meningeal inflammation may not be a necessary trigger for LME in MS, either.

The finding of PDE also requires more study and histopathologic validation. We chose the term paravascular (rather than perivascular) for consistency with the same finding described by Wu et al [25], who postulated that this finding may be due to an enlargement of the potential space between vascular walls and the glia limitans [22]. This concept is more in line with the visual appearance, as the regions of enhancement along vasculature are too large and amorphous to be exclusive to the vessel wall itself. Alternately, PDE also appears to occur in a similar pattern and location to CNS lymphatic vessels, and thus may represent reabsorption of gadolinium that has been deposited into the CSF from regions of permeable BBB (i.e. choroid plexus, dura). Other reports link PDE with alterations in glymphatic solute clearance [25, 37]. Although time dynamics for clearance cannot be accurately estimated based on the sequences obtained, the fact that the absolute difference between MS and HCs for PDE (and total enhancement) volume was much larger for Gd+ Early 7T FLAIR than for Gd+ Delayed 7T FLAIR suggests that rather than absolute measurements at any time point, perhaps time dynamics of contrast extravasation may be a more pathologic biomarker of meningeal vascular BBB permeability. This will need to be evaluated in future research.

Our finding that adding paravascular foci to the traditional definition of LME in MS studies [6] resulted in finding significant differences between MS and HCs for LMPE proportion and foci counts also warrants further exploration. The exclusion of paravascular foci from previous LME definitions may not have been physiologically warranted, as cortical vessels, including both veins and arteries, pass through the subarachnoid space for large portions of their course [22], and thus should also be susceptible to the meningeal inflammatory pathologic processes that are hypothesized to be behind LME. Further, the distribution of contrast around many of these foci spreads well beyond vascular wall borders and often appears to directly abut the cortical surface (examples in Fig 3). In fact, it is quite possible that the physiology of nodular and spread/fill pattern LME is the same as paravascular, with the only difference being the size of the blood vessel, with the central flow void in the former being obscured due to small size and partial volume averaging effects.

Other factors need to be considered when viewing the results of this study. Although no clear associations between LME or PDE and disability scores were found, we found a significant association between LME and age, and a similar non-significant trend was seen for PDE. This is consistent with our prior reports and with other reports of both LME and PDE in normal aging [13, 33, 38]. Evaluations of the glymphatic clearance of gadolinium through intravenous and intrathecal gadolinium injections show increased post-contrast FLAIR signal in brain parenchyma, CSF, and meningeal lymphatics with age, all suggesting alterations in the dynamics of waste clearance with age play a significant role in this imaging finding [9, 38]. Prior associations noted between LME in MS and disability levels and cortical atrophy, which also correlate with age, may also have been confounded by the impact of normal aging. Thus, the clinical relevance of LME should be called into question and all future analyses of LME in MS should take age into account.

The effect of DMT also needs to be considered. It is possible that in the course of the natural history of MS that meningeal inflammation resulting in meningeal BBB breakdown is indeed related to LME, but this effect is then blunted by the use of DMT. Most participants with MS in this study were on DMT, many of whom were on highly effective DMTs. One prior paper suggests the possibility that LME can be altered by highly effective DMTs [39]. However, this effect has not been seen in multiple other studies [40–42], nor was it replicated in this cohort. We also have recently reported the results of a clinical trial of ocrelizumab for modulation of LME and PDE, which showed no effect [19]. The high prevalence of both LME and PDE in multiple cohorts on DMT suggest that DMT may not impact ME to any significant degree. Regardless, the prevalence of highly effective DMT use and the relative stability of the cohort should be considered when accounting for differences seen between this and other similar studies in MS.

The potential contribution of manufacturer (Philips versus Siemens) differences in 3D-FLAIR acquisition methods to the 7T versus 3T findings in our study should be considered, as vendor-specific differences in LME detection have been previously described [28]. Thus, our findings may need to be replicated on a larger scale with multiple scanner types before attaining widespread applicability. Another possible confound to our findings may be the difference in gadolinium agents used between the 3T (gadobutrol) and 7T (gadoteridol) protocols. This choice was limited by local research scanner requirements. We cannot eliminate the possibility that the difference between the two agents may have contributed to the difference seen between 3T and 7T. It is unlikely this contributed a significant effect, however, as both agents are macrocyclic in nature and have extremely similar distribution half-lives (gadoteridol = 0.1 hours, gadobutrol = 0.2 hours) and elimination half-lives (gadoteridol = 1.5 hours, gadobutrol = 1.8 hours) [43]. Additional confounds may occur due to the slight difference in timing of sequence initialization after contrast administration in the two delayed

protocols (Gd+ Delayed 3T FLAIR = 21 minutes, Gd+ Delayed 7T FLAIR = 23 minutes), which may have influenced comparisons between these two protocols. However, review of tissue concentration curves after bolus injections of various gadolinium agents show that the steepest increases in tissue concentration are seen during the first 20 minutes, after which the minute-by-minute change in concentration change is quite small [18]. Thus, it is unlikely that a 2 minute difference had a profound impact on our conclusions. The lack of an early onset 3T FLAIR acquisition after administration of contrast for comparison also weakens our conclusions, as timing comparisons were only performed for the 7T protocols. While the timing comparisons between the two 7T protocols can be extrapolated to other field strengths, a truly comprehensive analysis would require a similar comparison at 3T. The relatively small sample size of HCs compared to the MS group may also be viewed as a limitation of our study. Many of the MS versus HC comparisons showed non-significant trends for differences in counts and volumes, particularly for LMPE and PDE–differences which may have been significant if the size of the MS and HC cohorts were more balanced. Finally, the registration method chosen may also have influenced the outcome of this study, as 3T images were reviewed after registration to the 7T space. It is unlikely this had significant impact, however, as all FLAIR images required transformation, including 7T FLAIR images into the 7T MP2RAGE space. Evaluation by the investigators of a few sample 3T scans with known enhancing foci revealed no difference in the visibility or intensity of enhancing foci before or after transformation. A more comprehensive evaluation of the impact of registration on enhancing foci is beyond the scope of this paper.

## Conclusion

Overall, our data suggests that future work may need to focus on the use of 7T MRI and adequate delays post-contrast administration for increased sensitivity, and that simple quantification of the number or volume of enhancing foci may not adequately quantify pathology. We hope that the data presented here spurs further investigation into whether evaluation of gadolinium-enhancement on FLAIR images is a useful biomarker for MS or not. It is also possible that confirmation of our results in the future may lead to conclusions that MCE is not a useful biomarker for MS and may be a manifestation of normal aging, rather than pathology. Either way, this work and future work to come will be an important contribution to the literature and provide a greater understanding of MS pathophysiology.

## Supporting information

**S1 Appendix.**
(DOCX)

**S1 Table. Meningeal enhancement subtypes and total enhancement.**
(DOCX)

**S2 Table. Demographic and clinical characteristics by presence of LMPE in the MS cohort.**
(DOCX)

**S3 Table. Comparison of rates of MCE by DMT class.**
(DOCX)

## Acknowledgments

We would like to acknowledge and thank the research staff and technicians at the Kennedy Krieger Institute at Johns Hopkins University and the University of Maryland for their

assistance in MRI acquisition. We would also like to acknowledge Kerry Naunton, Christina Ecker, and Christina Kingsley for their research coordinator work on this study.

## Author Contributions

**Conceptualization:** Daniel M. Harrison.

**Data curation:** Daniel M. Harrison, Yuxin Zeng, Seongjin Choi.

**Formal analysis:** Daniel M. Harrison, Yohance M. Allette, Seongjin Choi.

**Funding acquisition:** Daniel M. Harrison, Jun Hua.

**Investigation:** Daniel M. Harrison, Yohance M. Allette, Yuxin Zeng, Amanda Cohen, Shishir Dahal, Seongjin Choi, Jiachen Zhuo.

**Methodology:** Daniel M. Harrison, Jiachen Zhuo, Jun Hua.

**Project administration:** Daniel M. Harrison.

**Resources:** Daniel M. Harrison, Jiachen Zhuo, Jun Hua.

**Supervision:** Daniel M. Harrison.

**Validation:** Daniel M. Harrison.

**Visualization:** Daniel M. Harrison.

**Writing – original draft:** Daniel M. Harrison, Yohance M. Allette.

**Writing – review & editing:** Daniel M. Harrison, Yohance M. Allette, Yuxin Zeng, Amanda Cohen, Shishir Dahal, Seongjin Choi, Jiachen Zhuo, Jun Hua.

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
