## [Decision Letter · Decision Letter 0]

24 Apr 2024

PONE-D-24-06602Meningeal contrast enhancement in multiple sclerosis: assessment of field strength, acquisition delay, and clinical relevancePLOS ONE

Dear Dr. Harrison,

Thank you for submitting your manuscript to PLOS ONE. After careful consideration, we feel that it has merit but does not fully meet PLOS ONE’s publication criteria as it currently stands. Therefore, we invite you to submit a revised version of the manuscript that addresses the points raised during the review process. It is a pleasure to inform you that manuscript has received a sufficiently high priority to be published when appropriate revisions are made. The comments of the referee(s) are provided at the bottom of this letter to aid you in revising your paper. Please address the queries line-by-line in your 'Response Letter' document with specific details about any changes that were made in your revised manuscript, or the reasons why the suggested changes have not been made.

We look forward to receiving your revised manuscript.

Kind regards,

Arka Bhowmik, Ph.D.

Academic Editor

PLOS ONE

Journal Requirements:

"Data acquisition was funded by grants from the NIH (1R01NS104403-01) and Roche-Genentech. Funders had no direct control over any aspects of the study."

"Dr. Harrison has received research funding from EMD-Serono and Roche-Genentech and royalties/consulting fees from the American College of Physicians, Horizon Therapeutics, TG Therapeutics, EMD-Serono, and UpToDate Inc.

Drs. Allette, Zeng, Cohen, Dahal, Choi, Zhuo and Hua are without disclosures. "

Reviewers' comments:

Reviewer's Responses to Questions

**Comments to the Author**

1. Is the manuscript technically sound, and do the data support the conclusions?

Reviewer #1: Partly

Reviewer #2: Yes

2. Has the statistical analysis been performed appropriately and rigorously? 

Reviewer #1: I Don't Know

Reviewer #2: Yes

3. Have the authors made all data underlying the findings in their manuscript fully available?

Reviewer #1: Yes

Reviewer #2: Yes

4. Is the manuscript presented in an intelligible fashion and written in standard English?

Reviewer #1: Yes

Reviewer #2: Yes

5. Review Comments to the Author

Reviewer #1: 1. Recommendation for acceptance is conditional to adequate responses to specific suggestions.

2. While I believe the manuscript to be statistically sound, I defer final determination of statistical analysis to a reviewer/editor with a stronger basis in statistical assessment

5. I have provided a review/commentary on the subject written by me illustrating where the current manuscript fits in to current knowledge of the topic. This is provided for the editors in case they choose to use it. I reserve the possibility of making minor edits pending authors response to revision suggestions. I also separately provide specific comments and suggestions for authors to address in their manuscript.

Reviewer #2: This paper describes a cross-sectional analysis of leptomeningeal and dural contrast enhancement detected on 3T and 7T MRI compared to healthy controls. The study included 95 MS subjects and 17 controls. The study assessed the detection properties regarding the time of contrast administration. It is found that LME is more common in patients with MS and better detected in 7T scanners with postponed image acquisition.

The study is of high scientific value, because not so many research groups work on methodology of LME detection and scans' acquisition. Specifically, the use of 7 Tesla makes the study of great value due to high resolution of the scan and detection of different types of pathology. The results clearly state that postponed 7 Tesla imaging is more sensitive than the early one.

Several issues are worth to discuss:

1. A surprisingly high amount of LME detected in the control group. These findings are contradicting to another studies on 7 Tesla machine where HC group were positive in 6.7%. I would like the authors to discuss more on these results since they can lead to very forthcoming conclusions regarding the specificity of this biomarker. It is clearly stated, though, that LME may be found in another diseases, e.g. non-inflammatory CNS disorders, and even in healthy controls, so such a high percentage of LME and PDE in the control group would refer to eligibility criteria for controls and to review of their medical history.

2. Several factors may have had the impact of LME prevalence in this study and low difference to the HC group. For example, the a great amount of patients with MS in the study received anti-B-cell therapy and anti-VLA-4 therapy, that may decrease the prevalence of LME (although studies on that are still running). It is worth considering a therapy type or line (1st line and 2nd line) to include in regression models to account for.

Also, this patient group was rather moderately disabled, since the median EDSS score was 2.5. Some studies showed that patients of early stages can show no LME while it's accumulation appears on later stages. It would be interesting to see the numbers of LME in patients with different EDSS scores as a part of this research process. This may also be an interesting topic for further research.

3. Only few MRI Departments have access to 7 Tesla MRI, so, if this biomarker comes to be used routinely, a 3 Tesla will be the main machine used for it. I would like the authors to describe why they didn't use early time 3 Tesla sequence and only used delayed 3 Tesla acquisition. This may be of additional knowledge on the methodology.

It would be also of some importance if the authors would describe why specifically 21 and 23 minutes were selected as timepoints for MRI scans. Other labs use other timeframes and it would great if authors would describe their decision-making process.

Finally, I can conclude that this papers is of high scientific and practical value and requires only minor changes to make several issues more clear.

6. PLOS authors have the option to publish the peer review history of their article (what does this mean?). If published, this will include your full peer review and any attached files.

Reviewer #1: **Yes: **David S. Titelbaum, M.D.

Reviewer #2: **Yes: **Gleb Makshakov

---

## [Author Response · Author response to Decision Letter 0]

30 Apr 2024

All comments/revision suggestions have been addressed. Please see the attached response to reviewers document for details.

---

## [Decision Letter · Decision Letter 1]

13 May 2024

Meningeal contrast enhancement in multiple sclerosis: assessment of field strength, acquisition delay, and clinical relevance

PONE-D-24-06602R1

Dear Dr. Harrison,

We’re pleased to inform you that your manuscript has been judged scientifically suitable for publication and will be formally accepted for publication once it meets all outstanding technical requirements.

Kind regards,

Arka Bhowmik, Ph.D.

Academic Editor

PLOS ONE

Reviewers' comments:

Reviewer's Responses to Questions

**Comments to the Author**

1. If the authors have adequately addressed your comments raised in a previous round of review and you feel that this manuscript is now acceptable for publication, you may indicate that here to bypass the “Comments to the Author” section, enter your conflict of interest statement in the “Confidential to Editor” section, and submit your "Accept" recommendation.

Reviewer #1: All comments have been addressed

Reviewer #2: All comments have been addressed

2. Is the manuscript technically sound, and do the data support the conclusions?

Reviewer #1: Yes

Reviewer #2: Yes

3. Has the statistical analysis been performed appropriately and rigorously? 

Reviewer #1: I Don't Know

Reviewer #2: Yes

4. Have the authors made all data underlying the findings in their manuscript fully available?

Reviewer #1: Yes

Reviewer #2: Yes

5. Is the manuscript presented in an intelligible fashion and written in standard English?

Reviewer #1: Yes

Reviewer #2: Yes

6. Review Comments to the Author

Reviewer #1: (No Response)

Reviewer #2: I thank the authors of the paper for their work and suggest the paper to be accepted for publishing.

7. PLOS authors have the option to publish the peer review history of their article (what does this mean?). If published, this will include your full peer review and any attached files.

Reviewer #1: **Yes: **David S. Titelbaum, MD

Dept of Radiology

Shields Health Care

Brockton, MA USA

Reviewer #2: **Yes: **Gleb Makshakov

---

## [Editor Report · Acceptance letter]

16 May 2024

PONE-D-24-06602R1 

PLOS ONE

Dear Dr. Harrison, 

I'm pleased to inform you that your manuscript has been deemed suitable for publication in PLOS ONE. Congratulations! Your manuscript is now being handed over to our production team.

Kind regards, 

on behalf of

Dr. Arka Bhowmik 

Academic Editor

PLOS ONE